# Challenges implementing a carer support intervention within a national stroke organisation: findings from the process evaluation of the OSCARSS trial

Sarah Darley ,[1] Sarah Knowles,[1] Kate Woodward-Nutt,[2] Claire Mitchell,[3] Gunn Grande ,[4] Gail Ewing ,[5] Sarah Rhodes,[6] Audrey Bowen ,[3] Emma Patchwood [3]

► Prepublication history and supplemental material for this paper is available online. To view these files, please visit the journal online (http://dx.doi.org/10.1136/bmjopen-2020-038129).

► http://dx.doi.org/10.1136/bmjopen-2020-038777

**Correspondence to**
Dr Sarah Darley;
sarah.darley@manchester.ac.uk

## ABSTRACT

**Objectives** To examine the implementation of an intervention to support informal caregivers and to help understand findings from the Organising Support for Carers of Stroke Survivors (OSCARSS) cluster randomised controlled trial (cRCT).

**Design** Longitudinal process evaluation using mixed methods. Normalisation process theory informed data collection and provided a sensitising framework for analysis.

**Setting** Specialist stroke support services delivered primarily in the homes of informal carers of stroke survivors.

**Participants** OSCARSS cRCT participants including carers, staff, managers and senior leaders.

**Intervention** The Carer Support Needs Assessment Tool for Stroke (CSNAT-Stroke) intervention is a staff-facilitated, carer-led approach to help identify, prioritise and address support needs.

**Results** We conducted qualitative interviews with: OSCARSS cRCT carer participants (11 intervention, 10 control), staff (12 intervention, 8 control) and managers and senior leaders (11); and obtained 140 responses to an online staff survey over three separate time points. Both individual (carer/staff) and organisational factors impacted implementation of the CSNAT-Stroke intervention and how it was received by carers. We identified four themes: staff understanding, carer participation, implementation, and learning and support. Staff valued the idea of a structured approach to supporting carers, but key elements of the intervention were not routinely delivered. Carers did not necessarily identify as 'carers', which made it difficult for staff to engage them in the intervention. Despite organisational enthusiasm for OSCARSS, staff in the intervention arm perceived support and training for implementation of CSNAT-Stroke as delivered primarily by the research team, with few opportunities for shared learning across the organisation.

**Conclusions** We identified challenges across carer, staff and organisation levels that help explain the OSCARSS cRCT outcome. Ensuring training is translated into practice and ongoing organisational support would be required

## Strengths and limitations of this study

► Conducting a longitudinal process evaluation over the period of 3 years enabled us to ascertain how the intervention was implemented in practice and provide understanding of the neutral outcome of the trial.

► Mixed methods helped us capture the complexity of the intervention and allowed us to triangulate quantitative and qualitative data to increase validity and credibility of findings.

► We captured a range of perspectives from all participants in the trial, including staff implementing the intervention, carers receiving the intervention, senior managers and leaders within the organisation, and carers and staff within the control arm.

► Observation of implementation was not possible for practical reasons, which made intervention fidelity difficult to assess.

► We ran the process evaluation largely independent of the trial in order to establish effectiveness of the intervention under real world conditions. However, this meant that process information was not shared with the organisation until towards the end of the trial.

for full implementation of this type of intervention, with emphasis on the carer-led aspects, including supporting carer self-identification.

**Trial registration number** ISRCTN58414120.

## INTRODUCTION
### Background

Informal caregivers, such as family members, play a significant role in providing daily support to stroke survivors who experience a range of life-limiting difficulties.[1] The caring role can be rewarding[2] but is also challenging with significant health implications,[3] impacting on physical and emotional

well-being, financial security, professional and social activities.[4] Current support for carers differs according to location but generally comes from hospital stroke units, local authority, specialist carer organisations and stroke organisations.[5]

Supporting informal carers is a statutory priority[6–8] but there is little clarity on how best to identify and assess the needs of people caring informally for stroke survivors.[9–11] One relevant approach is the Carer Support Needs Assessment Tool (CSNAT) intervention[12]: a comprehensive evidence-based assessment tool integrated within a multistage person-centred approach to individualised support. It was developed, implemented and tested in the context of palliative care with positive outcomes.[13–15] As one of the few available evidence based interventions for carers that are carer-led and that enable comprehensive assessment and support, the CSNAT was considered worth exploring for supporting informal carers of people with long-term health conditions. For the present study, the CSNAT intervention was adapted for use in stroke care, through collaboration with a specialist stroke organisation and a group of lay carers. The involvement of carers in particular was integral in making the wording stroke specific and informing implementation of the intervention in practice.[16] The adaptation resulted in the CSNAT-Stroke intervention, including a staff training and implementation package tailored to the stroke organisation. Training was mainly delivered to front-line staff who are non-clinical support staff that provide personalised information, advice and support to stroke survivors and their carers. In brief, the intervention is a person-centred, structured process of needs assessment and support that is practitioner facilitated, but carer-led, comprising a four-stage process, including a needs assessment tool and action plan (see table 1).

Organising Support for Carers of Stroke Survivors (OSCARSS) was a cluster randomised controlled trial (cRCT) which aimed to investigate the clinical and cost-effectiveness of the CSNAT-Stroke intervention for supporting carers relative to a control of usual practice within the stroke organisation.[17] OSCARSS found no meaningful differences in self-reported clinical outcomes for carers, such as strain and distress, between intervention and control groups.[18] This paper presents findings from the embedded process evaluation of OSCARSS, which aimed to enrich understanding of the trial result, understand the factors influencing implementation and acceptability of a carer-led intervention, and consider implications for future research and practice.

## METHODOLOGICAL APPROACH

Process evaluations provide insight into the 'black box' of implementation of a complex intervention[19] and help explain discrepancies between expected and observed outcomes within a context.[20] The process evaluation for OSCARSS used longitudinal mixed methods to examine ongoing experiences of and perspectives on both usual

**Table 1**  CSNAT-Stroke intervention as intended in OSCARSS

| Item | CSNAT-Stroke intervention, as intended for implementation within the stroke organisation |
|---|---|
| Why | To provide a person-centred process of assessment and support for carers that is practitioner-facilitated but carer-led and tailored to the carer's individual needs, which are likely to change over time. The intervention assumes that carers may have difficulty considering and expressing their needs. |
| What (materials and procedures) | Materials/procedures include: (A) four-stage process (components outlined below), (B) a needs assessment tool, (C) an action plan. |
| | 1. Identifying and introducing: |
| | Staff identify the carer early and make clear that support is available ('scripts' for sensitive use of the term 'carer'). Staff introduce needs assessment tool to carers during separate time with the carer, providing an opportunity to discuss support needs. |
| | 2. Needs assessment |
| | Carers are given time to consider their support needs, self-completing the tool, identifying the areas in which they need more support and prioritising those most important to them. Staff normalise the practice of having separate time with the carer as well as stroke survivor if present, to support each as individuals. |
| | 3. Assessment conversation and tailoring |
| | An assessment conversation between carer and stroke practitioners identifies the carer's individual support needs and what they feel would be most supportive within the domains of the needs assessment tool that they have prioritised. Support may be directly delivered by staff at this time (eg, active listening, information, signposting/referrals) but helping carers identify sources of support they may wish to access themselves (self-help) or via family members and friends is also encouraged. Staff create service directories to support signposting/referral. |
| | 4. Shared plan for action and review |
| | Carers record a plan of supportive input describing the actions taken or to be taken by the practitioner or carer to address identified needs, which will be subsequently reviewed as appropriate. |
| | Each stage should be staff facilitated but carer-led. At all stages the carer should be given the opportunity to say what is most important to them and what they feel would help support them. |
| Who provided | Front-line staff: Essential criteria for recruitment to this role includes GCSE education, experience providing care to vulnerable people, and good communication skills. Training in stroke and stroke-specific care is provided by the organisation. Additional intervention specific training provided to front-line staff in clusters allocated to intervention. |
| | Training, codelivered by the service provider and the research team, is with groups of staff over a half-day session involving instructional videos and scripts, role-play, and workbook completion. |
| How | After training the intervention should be implemented by staff during their routine support visits. |
| Where | Typically in carers' homes. |
| When and how much | The intervention should be used every time a staff member has contact with a carer and requires a minimum of one face-to-face support contact with the carer, with reviews likely. |

Continued

| | Table 1 | Continued |
|---|---|---|
| **Item** | | **CSNAT-Stroke intervention, as intended for implementation within the stroke organisation** |
| Tailoring and modifications | | Staff training is modified for those joining after the primary roll-out of the intervention and adapted to one-to-one delivery by the research team using video conferencing. |

Key details of the intervention using descriptors from the TIDieR checklist,[38] adapted from Patchwood *et al*.[18]
CSNAT, Carer Support Needs Assessment Tool; GCSE, General Certificate of Secondary Education; OSCARSS, Organising Support for Carers of Stroke Survivors; TIDieR, Template for Intervention Description and Replication.

practice and the CSNAT-Stroke intervention from staff within the stroke organisation, as well as carers who receive their support and consented to join OSCARSS.

Normalisation process theory (NPT) was used as a framework to guide data collection and analysis, and sensitised us to certain points in the data for both agreement and resistance.[21] NPT is a commonly used framework in process evaluations of complex interventions in healthcare because it increases understanding of the dynamic processes involved during their implementation and helps to explain mechanisms that affect their outcomes.[22] NPT considers both agency (individual factors) and context (organisational factors) and proposes four constructs to explain how interventions are embedded into practice. Constructs include: Coherence (sense making), Cognitive Participation (engagement and enrolment), Collective Action (how the work happens in existing systems) and Reflexive Monitoring (evaluation).[23]

## SETTING AND SAMPLE

The OSCARSS cRCT ran from January 2017 to December 2018. It recruited 414 carers over an 18-month recruitment period from 35 randomised clusters made up of the stroke organisation's services across England and Northern Ireland (18 intervention and 17 control). Approximately 100 front-line staff who were employed at participating clusters during the study period were trained to participate in OSCARSS.

The process evaluation purposefully sampled a wide range of participating front-line cluster staff and consented carers from both trial arms. Staff, managers and senior leaders were approached to participate by email; carers were approached by telephone if they consented to be contacted for this purpose during initial trial recruitment.

## METHODS

We used mixed methods including training observations, qualitative interviews and online questionnaires at different time points to capture the perspectives of different stakeholders during the course of the study (figure 1). Triangulating methods and incorporating multiple perspectives and experiences aimed to increase

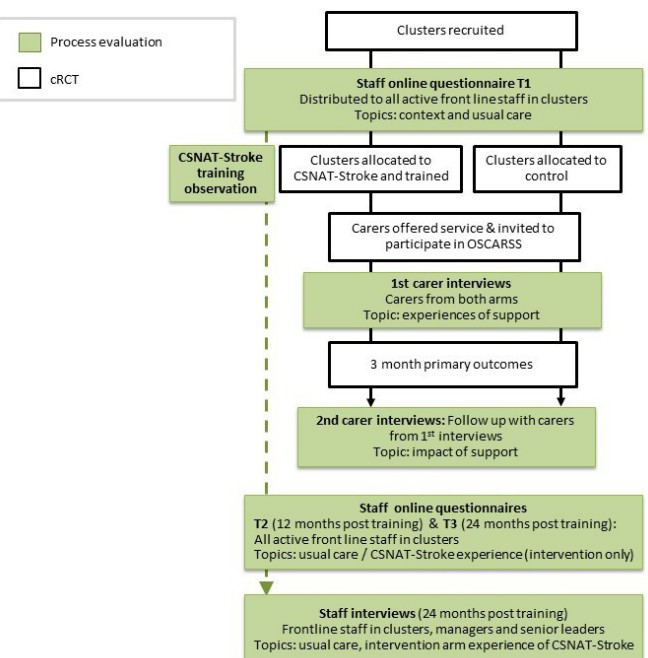

**Figure 1** Process evaluation methods. This figure shows how the mixed methods used within the process evaluation were conducted alongside the cRCT. cRCT, cluster randomised controlled trial; CSNAT, Carer Support Needs Assessment Tool; OSCARSS, Organising Support for Carers of Stroke Survivors.

the credibility and trustworthiness of our findings.[24] All participants gave informed consent.

Qualitative semistructured interviews were conducted to provide rich descriptions of experiences and perspectives of both carers and staff. We conducted telephone interviews with carers as soon as possible after joining the main trial. Carer interviews were completed by members of the trial team (EP and KW-N) and an experienced qualitative researcher (CM) between May 2017 and June 2018. Each carer was interviewed at the point of study consent and, if they agreed, again after primary outcomes had been collected at 3 months. Topic guides (online supplemental material 1) were informed by the research aim and feedback from the Research User Group (RUG), and explored carers' experiences of support and their perceptions of the impact of support received.

Staff interviews were conducted with cluster front-line staff, managers and senior leaders between August and November 2018. All staff interviews were conducted by SD who had not been involved with staff training or support. Staff interviews were conducted face to face or over the telephone, depending on staff location and preference. Interview topic guides (online supplemental material 2) were informed by NPT and aimed to capture context and carer support practices, and strategic organisational priorities (relevant for managers and senior leadership). For intervention-allocated front-line staff, additional questions explored changes to practice, mechanisms that supported changes and perceived outcome of changes to practice. Researchers made field notes

and reflections after all interviews, which lasted between 30 and 60 minutes. Interviews were audiorecorded with anonymised transcripts, field notes and reflections uploaded to NVivo software to facilitate thematic analysis.

Online questionnaires were distributed via email to all front-line staff involved in OSCARSS at three separate time points: T1 prerandomisation of clusters in August 2016, T2 and T3 postrandomisation in August 2017 and July 2018, respectively. Questionnaires asked about existing practice, attitudes towards supporting carers, the wider context within which staff worked, and demographic information. At T2 and T3, the questionnaire for intervention arm staff additionally included items based on the NOrmalisation MeAsure Development (NoMAD)[25] (online supplemental material 3). NoMAD contains Likert items designed to explore intervention implementation, based on NPT.

Staff training in the intervention arm was typically delivered by the stroke organisation over half day group sessions and observed by KWN and EP. Field notes were taken to capture: perceived engagement of attendees with the training; whether all components of the training had been delivered as intended; types of questions asked by attendees and responses given. Staff also completed training feedback forms to explore their understanding of key components of the intervention.

## Patient and public involvement

A study specific RUG of 10 individuals with experience of caring for a stroke survivor was set up in December 2015, at the planning stages of OSCARSS. Through regular meetings (2015–2019) and representation on the Trial Management Group the priorities, experiences and preferences of the RUG informed development of the research questions and the design, analysis and dissemination of both the trial and process evaluation. The RUG advised on participant recruitment and were central in limiting the burden of participation for carers. The RUG also supported adaptation of the CSNAT intervention and staff training package, including role-playing videos of the intervention in practice. A video summarising their role in OSCARSS is available on the study website (https://www.arc-gm.nihr.ac.uk/projects/oscarss) and a paper reflecting on their input.[16] In the early stages of data analysis, three meetings were held to focus on the process evaluation where SD, SK and EP shared and discussed themes from the data with the RUG. In these meetings, NPT and the value of this framework for understanding the findings were collaboratively discussed. The RUG helped us consider rival explanations, new ideas and implications on practice.

## DATA ANALYSIS
### Qualitative data

Several members of the research team (CM, EP, KW-N and SD) used an inductive approach to analyse the interview data. This involved repeated detailed readings of all transcripts to identify dominant and frequent aspects relevant to the research aim that were developed into a coding framework. This framework was further developed by process evaluation leads (SD and SK) who took a deductive approach to the analysis by applying the NPT framework to the findings. This approach enabled the existing codes to be organised under the four NPT constructs. Analysis was an iterative process and the combination of inductive and deductive approaches allowed codes to fit into a structure, relate to each other in meaningful, study important ways[26] and enabled identification of prominent themes. Other relevant data (eg, training materials, observation field notes and staff training feedback questionnaires) were reviewed throughout analysis (by SD and SK) to provide further context.

### Quantitative data

Questionnaire data were analysed descriptively to provide summary information on usual practice of supporting carers within the stroke organisation and included NoMAD items relating to use and acceptability of the intervention.

### Data synthesis

All data were synthesised and thematically analysed using the four main constructs of NPT[23 27] to identify patterns across the data.[28] As in previous studies this process allowed us to combine qualitative and quantitative findings[29] into one explanatory framework.[26] Researchers experienced in NPT (SD and SK) led this method of data analysis following a continuous, iterative process of reviewing and refining themes through discussions with the research team and RUG. We analysed at two levels: the individual and the organisation.

Draft findings were presented to multiple audiences, including internal PCPI collaborators (the RUG), Trial Management teams and at external conferences and seminars. This enabled input from a multiplicity of viewpoints and supported reflective discussions to reach consensus on core, prominent themes.[24]

## RESULTS
### Participants

Twenty-one carers of stroke survivors participated in first interviews (11 intervention, 10 control) and 11 agreed to follow-up interviews (4 intervention, 7 control) (table 2). Carers typically declined interviews when they felt they had nothing new to add.

Thirty-one staff members were interviewed (front-line staff: 12 intervention, 8 control; managers and senior leaders across clusters: 11). Thirteen additional front-line staff declined interviews. Staff participants were based across England and Northern Ireland, included both full and part-time workers and length of service ranged from less than a year to 23 years.

We observed 25 hours of staff training on the CSNAT-Stroke intervention over six group sessions, including 47

**Table 2** Carer characteristics

| Allocation | Gender | Age | Relation to stroke survivor | Stroke date | Ethnicity | SS needs score | Days from consent to interview | Status (1st/2nd interviews) |
|---|---|---|---|---|---|---|---|---|
| I | M | 50–59 | Partner | February 2017 | White | 2 | 33 | Completed 1 and 2 |
| I | F | 50–59 | Daughter | May 2017 | White | 2.7 | 28 | Completed 1 and 2 |
| I | M | 60–69 | Partner | May 2017 | Asian | 3 | 10 | Completed 1 and 2 |
| I | F | 60–69 | Partner | January 2017 | White | 2.7 | 18 | Completed 1 and 2 |
| I | F | 30–39 | Daughter | January 2017 | Asian | 1.7 | 4 | Opted out 2 |
| I | F | 50–59 | Daughter | August 2017 | White | 3 | 21 | Opted out 2 |
| I | F | 50–59 | Partner | June 2017 | White | 1 | 3 | Opted out 2 |
| I | F | 70–79 | Partner | March 2017 | White | 1.7 | 19 | Opted out 2 |
| I | M | 70–79 | Friend | March 2018 | White | 2 | 28 | Opted out 2 |
| I | M | 70–79 | Partner | June 2017 | White | 1 | 29 | Opted out 2 |
| I | F | 70–79 | Partner | November 2017 | White other | 1 | 65 | Opted out 2 |
| C | F | 40–49 | Partner | November 2017 | White | 1 | 16 | Completed 1 and 2 |
| C | F | 60–69 | Partner | January 2017 | White | 3 | 29 | Completed 1 and 2 |
| C | M | 60–69 | Partner | February 2017 | White | 1.8 | 20 | Completed 1 and 2 |
| C | M | 60–69 | Partner | November 2017 | White | 2.7 | 23 | Completed 1 and 2 |
| C | F | 60–69 | Partner | January 2018 | White | 1.8 | 17 | Completed 1 and 2 |
| C | F | 70–79 | Partner | February 2018 | White | 2.2 | 29 | Completed 1 and 2 |
| C | F | 70–79 | Partner | March 2017 | White | 3 | 14 | Completed 1 and 2 |
| C | F | 60–69 | Parent | July 2016 | White | 1.8 | 20 | Opted out 2 |
| C | F | 50–59 | Daughter | September 2017 | White | 2.7 | 77 | Opted out 2 |
| C | F | 50–59 | Daughter | December 2017 | White | 1.6 | 46 | Opted out 2 |

C, control arm; F, female; I, intervention arm; M, male; SS needs score, independence level as reported by Carer (Min score = 1, Max score = 3).

intervention-allocated staff. Response rates to the online staff questionnaire were: T1 72% (62 preintervention), T2 48% (20 intervention, 21 control), T3 35% (20 intervention, 17 control). Due to staff turnover, the same staff did not necessarily answer each questionnaire. Questionnaire respondents were predominantly female, aged 43 and over, worked part time and length of service ranged from less than 1 year to over 10.

**Themes**

In this section, we draw on four themes identified through the NPT framework to report factors that impacted on implementation of the intervention (table 3). For this purpose, we mainly present data from the staff intervention arm (interviews and questionnaires T2 and T3), as well as from carers, managers and senior leaders.

**Table 3** Themes identified through the NPT framework

| NPT construct and definition | Relevant questions considered to guide analysis | Theme from data |
|---|---|---|
| Coherence: Understanding and valuing of the intervention | Was the intervention understood as different from usual practice? <br><br> What were the beliefs and behaviours that defined the new approach? <br><br> Was there a shared understanding of the intervention? <br><br> Was there a benefit identified in following the intervention? | Limited understanding of different components of the intervention: Although all staff valued the ethos of a carer-led approach to supporting carers, despite the training staff demonstrated a limited understanding of the whole intervention. |
| Cognitive participation: Engagement in and activation of the intervention by individuals and groups | How was the intervention initiated? <br><br> Did staff and carers feel like legitimate participants in the intervention? <br><br> How were staff and carers enrolled and engaged in the intervention? <br><br> What promoted or inhibited participation in the intervention? | Limited carer participation in the intervention: Staff engaged with the intervention but struggled to persuade carers to participate and carers did not necessarily see themselves as legitimate participants. |
| Collective action: How participants work to carry out the intervention | How was the intervention carried out? <br><br> What were the beliefs and behaviours of staff and carers that define and organise the work? <br><br> How did staff fit the new approach into their existing practice and context? <br><br> Who was best placed to make use of the intervention? | The intervention was not fully implemented: Key components of the approach (such as a separate conversation between carer and staff member, and completion of the action plan) were not always implemented. |
| Reflexive monitoring: How the intervention is appraised and evaluated | What factors promoted or inhibited appraisal of the intervention? <br><br> How was the intervention monitored and evaluated? <br><br> How did staff and carers appraise the practice? <br><br> Was the intervention modified or adapted? | Limited opportunities to share learning and ongoing support for the intervention: All staff positively appraised the intervention as they understood it, but there were limited opportunities to share learning and receive ongoing support from across the stroke organisation. |

NPT, normalisation process theory.

We present our themes below, which encompass both the individual staff and carer level and the organisational level. Using the NPT framework to take both levels into account helps identify the individual behavioural changes and agency, as well as the organisational structural and cultural changes required to implement a complex intervention.[22]

### Understanding of the intervention

More than half of intervention arm staff respondents to T2 and T3 questionnaires said they could see how the intervention differed from their usual way of working and could be easily integrated into current practice. However, in the interviews where this issue was explored in detail there were mixed perceptions of the difference, particularly for staff who had been in the role for a number of years:

I'd say it was quite similar, it's just the way it's outlaid isn't it really? What I do is not on paper as such… I've got experience behind me and I've worked with carers for a long time. [Frontline staff member 2, intervention, interview]

The desire for a structured, formalised approach to offering carer support came through strongly from all staff, including senior leaders. Over half the intervention respondents to T2 and T3 agreed that staff in their service had a shared understanding of the intervention.

However, the intervention was primarily perceived as solely the paper-based needs assessment tool:

It's been a real pleasure to use, I consider it useful … in the beginning it was like okay great, more paper, but actually it's one of my most useful pieces of paper and I've got a lot of paper. [Frontline staff member 3, intervention, interview]

The interviews helped identify the lack of Coherence of the intervention through descriptions by staff of not fully implementing it and the similar accounts of support received by carers in both trial arms. Staff also appeared to see certain parts of the intervention, such as the action plan, as duplicating existing practice rather than a route to empowering carers to take ownership of actions for themselves:

I have to confess, I didn't [use the action plan]… I would follow it up in a letter … I don't see the point in doing [the action plan] …I'd be duplicating it. [Frontline staff member 4, intervention, interview]

Front-line staff positively appraised the OSCARSS training and, in feedback forms, highly endorsed that they understood each stage of the intervention and felt confident to deliver it. Despite this, interviews suggested that the training and ongoing support had not been sufficient to equip staff with a full understanding of the carer-led process of the intervention.

## Carer participation

Carers in both arms of the study reported doing a wide range of caring tasks including personal care, practical tasks, and financial administration. They described the impact of caring on their emotional and physical well-being, restricted social life, having to give up work, and financial implications:

> I'm 77 and it is hard work… it doesn't switch off. [Carer 1, intervention, interview 1]

However, during support visits carers did not always feel able to consider their own needs as their priority was the person they were caring for. This inhibited their Cognitive Participation in the intervention. Front-line staff saw this as a persistent issue, despite training for the intervention that included how to introduce CSNAT-Stroke during initial contact in a way that was sensitive to carer self-identification:

> Where there's sometimes debate is when you're labelling someone as a carer and often someone's wife, for example, would say—I'm not his carer, I'm his wife. [Frontline staff member 1, intervention, interview]

Carers in the control arm appraised the support they received from the stroke organisation just as positively as carers did in the intervention arm, particularly when they received little support from other organisations. Generally all carers interviewed said that they were grateful to have somebody interested in their needs and able to provide practical help.

## Implementation of intervention

The initiation of a separate conversation with a carer to discuss their needs is an important part of the intervention that was championed by the RUG. Observation notes showed that this point was often debated in the group training sessions, with some staff anticipating difficulty in talking with carers separately to stroke survivors. Staff described that historically these separate conversations tended to be intermittent and initiated by carers (often on the doorstep at the end of a visit), whereas OSCARSS training encouraged staff to routinely initiate a separate conversation and normalise this in advance of a visit. While staff saw the value of this in principle, they did not always initiate it in practice and instead many reported asking carers for the first time about meeting separately during the visit and in front of the stroke survivor:

> What I find is I'll introduce it but people don't want to go off and talk on their own. They want to stay with the stroke survivor. Some people do, but some people don't. They said, oh no, that's fine, we can talk quite openly with each other, so you can't make people go and talk on their own. [Frontline staff member 5, intervention. interview]

The lack of full implementation through Collective Action was also evident in the needs assessment process. Staff would often assume what carers' needs might be and anticipate not being able to meet them, which led to a reluctance to comprehensively discuss needs:

> To sort of say, okay, I recognise you're struggling with that and I don't have a solution… you're perceiving them to think, what was the point of asking me if I needed help - you've just highlighted the fact I need help and there's no services to help me. [Frontline staff member 1, intervention, interview]

Front-line staff had a well-meaning, strong desire to practically meet carer needs and 'fix' problems. However, core tenets of the carer-led approach of the intervention were that needs should be heard to be normalised and permit collaborative problem-solving, tailoring support to the carer's priorities.

Despite the lack of full implementation, components of the intervention that were understood were positively appraised by individual staff. For example, many expressed a desire to continue using the needs assessment tool in their meetings with carers beyond the trial. Managers were also keen to continue a systematic, structured approach to supporting carers.

## Shared learning and ongoing support

The focus on carers and the carer-led approach of the intervention aligned with organisational values and was strongly supported by senior leaders and managers. However, managers did not have a comprehensive understanding of the intervention and while front-line staff perception of management support for the intervention increased over time, they saw training for and communications about the intervention as coming from the research team rather than the organisation:

> I don't think really the [organisation] got involved much 'cause it was separate, wasn't it? Some [services] were included, some weren't, so I think they just left [the research team] to it and let you support us. [Frontline staff member 2, intervention, interview]

Front-line staff were viewed as most appropriate to deliver the intervention, although over half of T2 and T3 respondents in the intervention arm were neutral on whether work was assigned to those with skills appropriate to the intervention. At T2 and T3 front-line staff expressed openness to working with colleagues in new ways to use the intervention and in interviews front-line staff expressed a desire to learn from or share experiences with other teams.

However, there appeared to be few opportunities for shared learning on the intervention. Staff usually worked from home and did not have regular contact with each other, so opportunities were unlikely to happen without managerial planning. Reflexive monitoring was limited with no systematic monitoring or evaluation of the intervention within the organisation, although almost all of T2 and T3 intervention respondents agreed that feedback could be used to improve the intervention.

During OSCARSS, services across the organisation experienced significant reorganisation internal staffing changes and commissioning changes. In interviews some front-line staff described difficulty participating in the intervention within the context of internal and external changes and pressures:

> OSCARSS hasn't been a priority at all, we've been going through a lot of changes, we've lost some of our funding, we've had commissioned services being cut. There has been much more priority than doing the trial. [Frontline staff member 6, intervention, interview]

Just over half T2 and T3 intervention respondents agreed or strongly agreed that sufficient resources were available to support the intervention. However, in staff interviews a lack of time for and during visits to stroke survivors and carers was highlighted as inhibiting participation in the intervention:

> I think time would be the main thing, because… a lot of [frontline staff] just wouldn't necessarily factor the time in to a visit because they're already quite long, and there is already quite a lot of things to do within that visit. [Senior Leader, interview]

## DISCUSSION

The OSCARSS process evaluation helps us understand the context in which the intervention was delivered within the trial and provides some explanation of why no meaningful added benefit was evident when comparing intervention with usual practice. We identified significant challenges to successful implementation of the intervention at both individual and organisational levels. Front-line staff were mainly positive about the intervention, insofar as they wished to enhance carer support, but they did not have a full understanding of the carer-led aspects of the intervention. As in previous studies[30] this lack of coherence was a key factor in the intervention not being fully implemented. The training that front-line staff received through OSCARSS (half a day face to face or 1–3 hours over Skype) and ongoing informal support from the research team was not sufficient to provide this understanding. Front-line staff cared deeply about supporting carers, but their tendency to anticipate and 'fix' needs limited opportunities for the approach to be carer-led. Key stages of the intervention were also not routinely implemented, such as a separate conversation and completion of an action plan. Consistent with this finding, interviews with carers demonstrated little difference in support provided across the trial's intervention and control arms. Despite positive engagement and organisational enthusiasm for OSCARSS, the intervention was not delivered as intended. The findings demonstrate challenges to delivery of carer-led assessment and support, including overcoming the reluctance of carers to express their own needs, front-line staff anticipating carer needs, and the importance of ongoing organisational support and learning.

Although there is strong evidence for the need to support carers of stroke survivors[31 32] and a statutory requirement to do so[8] those considering implementing carer-led interventions should be aware of the challenges and solutions highlighted by OSCARSS. Carers in our study described the range of tasks they were responsible for in providing care and the impact this has on their lives. However, in line with other studies the carers did not always feel ready to accept support for themselves and did not necessarily see themselves as 'carers'[33 34] and the intervention was not implemented in a way to address this issue. Using NPT emphasised that motivation and 'buy in' for an intervention relies on engagement of both practitioner and service recipient to 'enrol' themselves as eligible and appropriate for an intervention to work. This may be particularly challenging for carers. Our findings contribute to this often overlooked yet vital aspect in implementation theory[35] and our multi-takeholder analysis enabled us to explore both individual and wider contextual factors that impact on implementation in practice.

Our process evaluation has shown the difficulty implementing a complex intervention that follows a carer-led ethos, despite strong organisational enthusiasm and staff willingness to support carers. Our findings show that staff would have welcomed additional ongoing support and shared learning opportunities to gain a full understanding of the carer-led approach. Support mechanisms used in previous CSNAT intervention implementations, such as developing staff champions[36 37] may have ensured that managerial commitment was communicated to front-line staff. However, this was difficult within the design of the cluster RCT as in some services a cluster may have only included one or two staff members. The trial team tried to offer similar support and championing to intervention arm front-line staff, but our findings indicate that such support would have been more effective coming from the organisation itself. Many carers across both trial arms said that front-line staff in the organisation were one of their main sources of support and this was highly valued. In OSCARSS, the intervention was developed to be used within the pragmatic organisational context of individual staff supporting both stroke survivor and carer, requiring no additional resources. However, our findings highlight the barriers for front-line staff to fully support both individuals within their available resources and limited time. Our findings also highlight the particular complexity of meeting needs of carers who struggle to recognise their own needs, consideration of timing of support, and how to sensitively and appropriately facilitate conversations with carers and stroke survivors.

The process evaluation was designed and delivered by a dedicated evaluation team (SD and SK) alongside the main trial and used multiple methods over time to generate rich and in-depth data. It used NPT to bring together these data that provided a range of perspectives at different time points and sampled from both trial arms across clusters. Through examining implementation at the individual and

organisational level, the process evaluation offers insight into why the intervention was not fully implemented and contributes to knowledge on implementing carer-led approaches in healthcare. A limitation to the study was that observation of the intervention in real-time was not conducted. Reasons for this included the difficulty of minimising Hawthorne effects on staff implementing the intervention, ethical considerations of carers discussing sensitive information at a potentially distressing time, and practicalities of observation within a large-scale study of 35 clusters. This limitation impacted on the extent to which we could assess fidelity as we were not able to triangulate interview and question data with observations of practice. However, we recruited carers and staff from different geographical areas within the trial and triangulated multiple data to get an understanding of implementation from the perspective of several stakeholder groups. The process evaluation was also run largely independent of the trial in order to establish effectiveness of the intervention under real world conditions.[20] However, this resulted in sharing process information with the organisation only towards the end of the trial, meaning that additional strategies to help support full implementation of the intervention were not explored. In hindsight, taking into account our findings, it may have been useful to have planned the process evaluation in stages with an early review, and also run an initial feasibility study before the full trial.

The process evaluation showed that the intervention was not fully implemented as intended, therefore we cannot say whether it may provide a benefit to carers of stroke survivors. Using NPT we have highlighted areas that need to be addressed in future research of this intervention or similar person-led approaches to supporting carers of stroke survivors. Considerations include first, how to train and support staff including strategies to help with engaging carers reluctant to prioritise their own needs. Second, providing clear communication pathways to ensure adoption of the intervention, involving management goals top down to front-line staff and also bottom up enabling practitioners to raise implementation difficulties and have them resolved. Third, having a network of staff champions as role models and for problem solving. Fourth, clarity at the outset about resources needed for implementation and sustainability.

## CONCLUSIONS

This paper shows the value of process evaluation in helping to interpret neutral trial results, such as OSCARSS. It demonstrates how both staff and service user perceptions and organisational demands can impede the delivery of key intervention components, even with significant buy-in to the need for the intervention itself. Our findings identified certain challenges, as well as opportunities, that contribute to knowledge on implementing carer-led support interventions. We recommend that future intervention work should concentrate on addressing organisational factors which can both help and hinder implementation and that all levels within the organisation are engaged, including undertaking training, to have a full understanding of all components of the intervention.

**Author affiliations**
[1]Alliance Manchester Business School, The University of Manchester, Manchester, UK
[2]NIHR Collaboration for Leadership in Applied Health Research and Care, Greater Manchester, Manchester, UK
[3]Division of Neuroscience and Experimental Psychology, The University of Manchester, Manchester Academic Health Sciences Centre (MAHSC), Manchester, UK
[4]Division of Nursing, Midwifery and Social Work, School of Health Sciences, The University of Manchester, Manchester Academic Health Sciences Centre (MAHSC), Manchester, UK
[5]Centre for Family Research, University of Cambridge, Cambridge, UK
[6]Centre for Biostatistics, Division of Population Health, Health Services Research and Primary Care, The University of Manchester, Manchester Academic Health Sciences Centre (MAHSC), Manchester, UK

**Acknowledgements** The authors would like to thank all of the carers and staff members who participated and contributed to this study. Jessica Stolberg contributed to collection of questionnaire data and Verity Longley contributed to analysis. The trial steering committee contributed to the study design and interpretation of findings. The study specific carer advisory research group contributed to the study design and interpretation of findings: Kelly Burke, Christine Halford, Natalie Halford, Geoff Heathcote, Kath Purcell and Ben Wright. We would like to thank members of the supportive NIHR CLAHRC GM team who helped make the study possible including: Caroline O'Donnell (data analyst); Alison Littlewood and Katy Rothwell (programme managers); Sam Wilkinson, Amy Woodhouse and Rose Crees (administrative assistants); Aneela Macavoy and Zoe Ashton (research facilitators). We also wish to thank and acknowledge Stroke Association staff, including Chris Larkin, Elaine Roberts, Cass Saunders and Kathryn Staley.

**Contributors** All authors (SD, SK, KW-N, CM, GG, GE, SR, AB and EP) contributed to the design of this study. SD, KW-N, CM and EP contributed to recruitment, collected the data and contributed to the analysis. All authors (SD, SK, KW-N, CM, GG, GE, SR, AB, EP) contributed to interpretation of findings and writing of the manuscript.

**Funding** This project was supported by the National Institute for Health Research Collaboration for Leadership in Applied Health Research and Care (NIHR CLAHRC) Greater Manchester, grant number (N/A) and Stroke Association, grant number (N/A). Some Stroke Association staff were research participants in this study and others contributed to discussions about findings and dissemination. The first author had full access to all the study data and had final responsibility for the decision to submit for publication.

**Disclaimer** NIHR had no role in study design, data collection, data analysis, data interpretation, or writing of the paper. Stroke Association partnered with NIHR in funding this study and was the specialist stroke service provider in OSCARSS. The views expressed in this article are those of the author(s) and not necessarily those of the NIHR, the Department of Health and Social Care, or the Stroke Association.

**Competing interests** GG reports grants from NIHR CLAHRC, during the conduct of the study; In addition, GG has a patent Copyright issued. GE reports grants from NIHR, during the conduct of the study; In addition, GE has a patent Copyright issued. SR reports grants from NIHR, during the conduct of the study. AB reports grants from NIHR, grants from Stroke Association, during the conduct of the study; grants from Stroke Association, grants from NIHR, outside the submitted work. EP reports grants from Stroke Association, outside the submitted work. SD, SK, KW-N and CM have nothing to disclose.

**Patient consent for publication** Not required.

**Ethics approval** Ethics approval was obtained from the North West Lancaster Research Ethics Committee (ref: 16/NW/0657) and University of Manchester (ref: AMBS-2016–22).

**Provenance and peer review** Not commissioned; externally peer reviewed.

**ORCID iDs**
Sarah Darley http://orcid.org/0000-0001-5420-6774
Gunn Grande http://orcid.org/0000-0003-2200-1680
Gail Ewing http://orcid.org/0000-0001-9547-7247
Audrey Bowen http://orcid.org/0000-0003-4075-1215
Emma Patchwood http://orcid.org/0000-0002-4198-5761

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
