## [Reviewer comments · BMJ Open]

ARTICLE DETAILS

TITLE (PROVISIONAL)	CHALLENGES IMPLEMENTING A CARER SUPPORT INTERVENTION WITHIN A NATIONAL STROKE ORGANISATION: FINDINGS FROM THE PROCESS EVALUATION OF THE OSCARSS TRIAL
AUTHORS	Darley, Sarah; Knowles, Sarah; Woodward-Nutt, Kate; Mitchell, Claire; Grande, Gunn; Ewing, Gail; Rhodes, Sarah; Bowen, Audrey; Patchwood, Emma

VERSION 1 – REVIEW

REVIEWER	Dr Jessica Hall Bradford Institute for Health Research, UK
REVIEW RETURNED	07-Apr-2020

GENERAL COMMENTS	This research aimed to examine the implementation of an intervention to support informal caregivers and to help understand findings from the Organising Support for Carers of Stroke Survivors (OSCARSS) cluster randomised controlled trial (cRCT). This is a valuable piece of research which provides important learning for how carers of stroke survivors are best supported. However, the manuscript could be improved prior to being accepted for publication. Please could the authors consider the following points: Background Page 4, line 22 - where the authors refer to caring being rewarding it would be appropriate to reference Nan Greenwood's research: 'Positive experiences of caregiving in stroke: a systematic review' Page 4, lines 25 and 28 – please add references to support the key points made. Page 4, lines 27-28 – please provide some justification for why the CSNAT is relevant to this setting when it was originally used in palliative care. Page 4, lines 31 and 32- it is interesting that carers were involved in adapting the CSNAT, could further details be provided about this process Table 1 provides a very clear overview Methodological approach Page 5, lines 58-60 – please provide further justification for using normalisation process theory and how this was used as a sensitising framework. Page 6, line 30 – please provide examples of the 'frontline staff' involved Page 7, lines 13-14 - the ethics section feels a little out of place. Could it be moved to the start of this section?
---

	Page 7 - the data analysis section would be clearer if subheadings were used to divide the qualitative and quantitative analysis Page 7, lines 19- 26- the process used to analyse the data needs more clarification. The authors mention that data were analysed thematically but using what approach? Reference 20 appears to be a qualitative book so is too vague. As a reader it is difficult to know what the approach involved in lines 23-26. 'Induction-deduction' is an unusual phrase. It would be clearer to say a combination of inductive and deductive approaches allowed for... Themes Pages 9/10. The table is a nice way of presenting the themes to the reader and this findings section is very interesting. However, it is really difficult to understand how the themes outlined in the table fit with the narrative that follows. The focus on the individual and organisational levels feels a little out of the blue without providing more clarity. Could either the table or text be changed to create more consistency? Themes as subheadings would also help the reader to navigate through this section. Discussion The discussion is well written and includes some important messages. The authors may want to check for consistency if changes are made to the structure of the results section.
--	--

REVIEWER	Hueiming Liu The George Institute for Global Health, Australia
REVIEW RETURNED	11-May-2020

GENERAL COMMENTS	Thank you for this well written and interesting process evaluation of a cRCT investigating the effectiveness of a carer support intervention which had a neutral outcome. The authors described the rationale for the cRCT and the process evaluation clearly. This included an in-depth description of the complex intervention as intended, and the clear application of NPT as a sensitising framework in the methods and data analysis. The results provide the explanation that the intervention was not delivered as intended (not truly carer-led) due to difficulties in carers identifying their own needs, and also the organisational challenges for the staff in implementing the CSNAT intervention such as available resources and time. Overall the discussion is clear, with relevant and important implications. Some additional comments/thoughts for your kind consideration, in regards to potential implications for longitudinal process evaluations: As it was a longitudinal study, it may be good to have some additional insights, on hindsight, as to whether the need for additional implementation strategies such as champions, and clarity at the outset about resources needed for implementation and sustainability could potentially have been identified earlier e.g. during the presentation of draft findings to multiple audiences. (Pg7 Ln 32) In addition, it was interesting to see the NoMAD results at T2 and T3, which seemed to indicate that CSNAT was well received and different from usual care, but that it was only through the interviews with frontline staff and managers at 24 months post training that illustrated how the intervention was not delivered as intended. It made me wonder about the 'dissonance' of these results, so to speak. Would appreciate your reflections on that.
--

	What did the field notes about the observation of the training find? Did this indicate in part that more training may have been required? ----- Thanks again for this interesting paper. Best wishes.
--	--

VERSION 1 – AUTHOR RESPONSE

Reviewer	Revision	Section	Addressed
Reviewer: 1	Page 4, line 22 - where the authors refer to caring being rewarding it would be appropriate to reference Nan Greenwood's research: 'Positive experiences of caregiving in stroke: a systematic review'	Background	Reference added on to this section and in reference list: Mackenzie A, Greenwood N. Positive experiences of caregiving in stroke: a systematic review. Disabil Rehabil. 2012;34(17):1413-1422.
	Page 4, lines 25 and 28 – please add references to support the key points made.	Background	Added reference to Stroke: A Carers guide and also three references added regarding supporting carer needs.
	Page 4, lines 27-28 – please provide some justification for why the CSNAT is relevant to this setting when it was originally used in palliative care.	Background	Justification added to using the CSNAT intervention to support informal carers of long-term health conditions.
	Page 4, lines 31 and 32- it is interesting that carers were involved in adapting the CSNAT, could further details be provided about this process Table 1 provides a very clear overview	Background	Further details added to this section on the carers involvement in adapting the CSNAT and we have also added a reference to the published paper on the involvement of carers in the study.
	Page 5, lines 58-60 – please provide further justification for using normalisation process theory and how this was used as a sensitising framework.	Methodological approach	Added reason for using NPT and clarified its used as a sensitising framework

	Page 6, line 30 – please provide examples of the 'frontline staff' involved	Setting and sample	A description of what the frontline staff do has been provided. (their specific job title has not been included to avoid identification of participants)
	Page 7, lines 13-14 - the ethics section feels a little out of place. Could it be moved to the start of this section?	Methods	The ethics section has been moved to the end of the first paragraph in the Methods section.
	Page 7 - the data analysis section would be clearer if subheadings were used to divide the qualitative and quantitative analysis	Data analysis	Three headings have been added to make the analysis section clearer: qualitative data, quantitative data, and data synthesis.
	Page 7, lines 19- 26- the process used to analyse the data needs more clarification. The authors mention that data were analysed thematically but using what approach? Reference 20 appears to be a qualitative book so is too vague. As a reader it is difficult to know what the approach involved in lines 23-26.	Data analysis	Further detail has been added regarding thematic analysis with a more specific reference.
	'Induction-deduction' is an unusual phrase. It would be clearer to say a combination of inductive and deductive approaches allowed for...	Data analysis	Changed now to the suggested sentence
	Pages 9/10. The table is a nice way of presenting the themes to the reader and this findings section is very interesting. However, it is really difficult to understand how the themes outlined in the table fit with the narrative that follows. The focus on the individual and organisational levels feels a little out of the blue without providing more	Themes	We have changed the text immediately after the table to link the individual and organisational levels to the NPT themes. The themes have also been added to the subheadings of the findings section to provide further clarity.

	clarity. Could either the table or text be changed to create more consistency? Themes as subheadings would also help the reader to navigate through this section.		
	The discussion is well written and includes some important messages. The authors may want to check for consistency if changes are made to the structure of the results section.	Discussion	Discussion section has been updated to be consistent with revision changes.
Reviewer: 2	As it was a longitudinal study, it may be good to have some additional insights, on hindsight, as to whether the need for additional implementation strategies such as champions, and clarity at the outset about resources needed for implementation and sustainability could potentially have been identified earlier e.g. during the presentation of draft findings to multiple audiences. (Pg7 Ln 32)	Discussion	Included reflections on the design of the trial that made the use of champions and additional resources difficult and how the trial team did try and address this. Also added the methodological limitation of running the process evaluation largely separate to the trial so process information was only shared with the organisation towards the end of the trial (added in the Discussion and the Strengths and Limitations section).
	In addition, it was interesting to see the NoMAD results at T2 and T3, which seemed to indicate that CSNAT was well received and different from usual care, but that it was only through the interviews with frontline staff and managers at 24 months post training that illustrated how the intervention was not delivered as intended. It made me wonder about the 'dissonance' of these results, so to speak. Would appreciate your reflections on that.	Findings and discussion	We see this dissonance as coming from a lack of coherence about the intervention and how it was intended to be implemented. We have included how the interviews helped us identify this lack of coherence and have also added this point to the discussion section. In the Findings section, we have also included reference to the observation notes from training to support our findings on the difficulty staff felt about asking to speak with carers separately.

	What did the field notes about the observation of the training find? Did this indicate in part that more training may have been required?		
FORMATTING AMENDMENTS	Statements Please embed the following statement to your main document just before your reference list. a. Data sharing statement		Added the Data Sharing Statement before the reference list.
	Patient and Public Involvement statement: Kindly provide a separate 'Patient and Public Involvement' statement in the methods section of your manuscript to avoid confusion.		We have moved our PPI statement to the end of the Methods section. We have titled it 'PATIENT, CARER, AND PUBLIC INVOLVEMENT (PCPI)' as we feel the inclusion of carers is more relevant to our study but please feel free to change it to 'Patient and Public Involvement' if required.
	Please provide figure legend/caption Please include figure legends at the end of your main manuscript.		Figure legend has been added to the end of the main manuscript after the references.

VERSION 2 – REVIEW

REVIEWER	Dr Jessica Hall Academic Unit for Ageing and Stroke Research, Bradford Institute for Health Research, Bradford Royal Infirmary, Bradford, United Kingdom
REVIEW RETURNED	24-Jul-2020

GENERAL COMMENTS	Thank you to the authors for addressing the comments that were provided previously. The manuscript is much improved following the revisions. However, having reviewed this manuscript again, there are some points that the authors should consider before this is interesting paper is published: Setting and Sample heading: Within this subsection, the authors have provided an explanation about front line workers. Could this be moved up to table 1 where front line workers are referred to if the readers are going to see the table first. Qualitative data within methods: Under this section could the tense be changed to be more active and say something along these lines: 'Several members of the research team used an inductive approach to develop a coding framework. This framework was further developed by process evaluation leads (SD and SK) took a deductive approach to the analysis by applying the NPT framework to the findings.' It is not
---

	currently quite clear what actually happened for the analysis, so any further details that could clarify this would be beneficial. Table 3 within the results section: There still seems to be a mismatch between what is presented in the table and the text which follows. It is appreciated that the authors have attempted to clarify in the text below the table. However, to provide further clarity it would help to see the subheadings from the 'themes from data' column within the text underneath. Having broad headings of individual staff and carer level factors and organisational level and contextual factors makes it difficult to navigate as a reader. This would also better highlight the interesting findings.
--	--

REVIEWER	Hueiming Liu The George Institute For Global Health, University of New South Wales, Australia,
REVIEW RETURNED	14-Aug-2020

GENERAL COMMENTS	Thank you, the paper reads well and has addressed the comments adequately. Best wishes.
--

VERSION 2 – AUTHOR RESPONSE

Reviewer	Revision	Section	Addressed
Reviewer: 1	Within this subsection, the authors have provided an explanation about front line workers. Could this be moved up to table 1 where front line workers are referred to if the readers are going to see the table first.	Setting and Sample heading	The definition of frontline staff has been added to the end of the Background section so that readers will see this before Table 1.
	Under this section could the tense be changed to be more active and say something along these lines: 'Several members of the research team used an inductive approach to develop a coding framework. This framework was further developed by process evaluation leads (SD and SK) took a deductive approach to the analysis by applying the NPT framework to the findings.' It is not currently quite clear what actually happened for the analysis, so any further details that could clarify this would be beneficial.	Qualitative data within methods	The tense of this paragraph has been changed as suggested and an extra sentence added to provide further details on the analysis.

	There still seems to be a mismatch between what is presented in the table and the text which follows. It is appreciated that the authors have attempted to clarify in the text below the table. However, to provide further clarity it would help to see the subheadings from the 'themes from data' column within the text underneath. Having broad headings of individual staff and carer level factors and organisational level and contextual factors makes it difficult to navigate as a reader. This would also better highlight the interesting findings.	Results section	We've added the four headings from the themes in Table 3 to the Results section – we've shortened these headings to ensure they work as sub-headings but still keep the consistency of the themes (Understanding of the intervention, Carer participation, Implementation of intervention, and Shared learning and ongoing support). With the addition of these sub-headings we have removed the individual and organisational level headings, and also the NPT components in brackets. However the NPT components are still highlighted in bold in each section. We have ensured that the text follows the same order as in Table 3, so the 'Understanding of the intervention' heading is first. We've also edited the text before the subheadings (straight after Table 3) so the text flows better without the explicit headings of individual and organisation levels. Additionally we've included the four themes in the abstract results section.
Reviewer 2	Thank you, the paper reads well and has addressed the comments adequately.		Thank you very much for your time

We also made the following slight changes to improve readability of the text:

Section	Original	Changed
Data Analysis - Data synthesis	...supported reflective discussions to reach consensus on core, prominent themes	Reference added [24] on page 7.
Themes: Implementation of intervention	needs should be heard to	needs should be heard to be normalised – changed to avoid duplicate of 'needs'

	normalise needs	
Discussion	and a need for ongoing organisational support and learning.	and the importance of ongoing organisational support and learning - changed to avoid duplicate of 'needs'
	and the need to consider timing of support,	and the consideration of timing of support - changed to avoid duplicate of 'needs'
	The process evaluation was led by experienced researchers separate to the trial team	The process evaluation was designed and delivered by a dedicated evaluation team (SD, SK) alongside the trial – to make the relationship between the trial and process evaluation clearer
	all levels within the organisation are engaged in implementation, including undertaking training	all levels within the organisation are engaged, including undertaking training – deleted 'implemented' to avoid repetition.
	Secondly, ensuring clear communication pathways to ensure adoption of the intervention	Secondly, providing clear communication - replaced 'ensuring' to avoid duplication.

Following feedback from the editorial assistant we have also made changes to:

- the AUTHOR CONTRIBUTIONS section and moved anyone who is not a named author into the Acknowledgements section
- ensure that the Patient and Public involvement heading is as per the guidance
- add 'N/A' next to the funder in the funding section in the main document and in Scholar One

VERSION 3 – REVIEW

REVIEWER	Dr Jessica Hall Bradford Institute for Health Research, UK
REVIEW RETURNED	25-Nov-2020
GENERAL COMMENTS	Many thanks to the authors for taking the time to adequately address the suggested revisions. The paper reads well and provides a valuable contribution to this field of research.